# Multiple Perspectives Reveal the Role of DNA Damage Repair Genes in the Molecular Classification and Prognosis of Pancreatic Adenocarcinoma

**DOI:** 10.3390/ijms231810231

**Published:** 2022-09-06

**Authors:** Yujie Li, Ke Zhang, Linjia Peng, Lianyu Chen, Huifeng Gao, Hao Chen

**Affiliations:** 1Department of Integrative Oncology, Fudan University Shanghai Cancer Center, Shanghai 200032, China; 2Department of Oncology, Shanghai Medical College, Fudan University, Shanghai 200032, China

**Keywords:** pancreatic adenocarcinoma, DNA damage repair, molecular classification, prognosis, immune

## Abstract

Pancreatic adenocarcinoma (PAAD) is a highly heterogeneous and immunosuppressive cancer. This study investigated the diversity of DNA damage repair (DDR) and immune microenvironment in PAAD by transcriptomic and genomic analysis. Patients with PAAD were divided into two DDR-based subtypes with distinct prognosis and molecular characteristics. The differential expression genes were mostly enriched in DDR and immune-related pathways. In order to distinguish high- and low-risk groups clinically, a DDR- and immune-based 5-gene prognostic signature (termed DPRS) was established. Patients in the high-risk group had inferior prognosis, a low level of immune checkpoint gene expression and low sensitivity to DDR-associated inhibitors. Furthermore, single-cell sequencing was used to observe the performance of the DDR-based signature in a high dimension, and immunohistochemistry was used to verify the relationship between the genes we identified and the prognosis of patients with PAAD. In conclusion, the DDR heterogeneity of PAAD was demonstrated, and a novel DDR- and immune-based risk-scoring model was constructed, which indicated the feasibility of DPRS in predicting prognosis and drug response in PAAD patients.

## 1. Introduction

Pancreatic adenocarcinoma (PAAD) is a devastating tumor with an extremely poor prognosis, which was listed as the fourth leading cause of cancer-related death [1]. Due to the insidious onset and metastatic nature of PAAD, more than 50% of patients miss the opportunity with an advanced stage at the first diagnosis [2]. Some studies have proposed molecular typing of PAAD based on genomic, transcriptome and metabolome data, and have broadened the understanding of the molecular phenotype of PAAD and provided effective targeted therapy options [3,4,5]. However, the molecular mechanism of poor prognosis in PAAD remains unclear.

DNA damage, which threatens the integrity of the genome, is a potential cause of cancer. Cells have developed a variety of mechanisms for detecting, sending signals and repairing DNA damage, and these pathways are collectively known as DNA damage repair (DDR) [6]. The emergence of targeted DDR inhibitors has shown initial success in patients with advanced platinum-sensitive PAAD with DDR gene alterations or BRCA mutations [7,8]. Recently, DDR deficiency has emerged as one of the important determinants of tumor immunogenicity, and there is increasing evidence to support the concept that DDR-targeted therapies can increase anti-tumor immune responses [9]. However, transcriptome and genomic analysis of PAAD from the perspective of DDR gene heterogeneity remains limited.

Here, we aim to analyze the transcription profile of the DDR gene in the PAAD comprehensively. Two DDR gene subtypes with different clinical and immunological characteristics were identified. A five-gene signature based on DDR and immune-related genes that can successfully differentiate between high-and low-risk groups of PAAD patients was developed, providing an alternative method for predicting the response to DDR-targeted inhibitors and immune checkpoint inhibitors (ICI). Our data revealed multiple aspects of DDR alterations in PAAD that may be useful in guiding therapy and prognostic monitoring.

## 2. Results

A schematic presentation of the research procedure is shown in Figure 1.

### 2.1. Superior Overall Survival (OS) Was Characterized in DDR-Subtype 1

To reveal the heterogeneity of DDR genes in PAAD, all 168 patients from The Cancer Genome Atlas (TCGA) were assigned to different subtypes based on 276 DDR gene expression profiles. Considering the unsupervised clustering results and clinical significance, two DDR subgroups were identified (Figure 2A–C). Cluster1 (*n* = 122) was designated as DDR-subtype1, and cluster2 (*n* = 46) was designated as DDR-subtype2. Most of the DDR genes in DDR-subtype1 were down-regulated, and in DDR-subtype2 they were up-regulated (Figure 2D). There was a significant difference in OS between the two subtypes (Figure 2E). The OS of DDR-subtype2 was inferior to that of subtype1 (*p* < 0.001). This prompted us to continue to explore the relationship between DDR and prognosis in patients with PAAD by exploring the molecular characteristics of these two DDR subtypes.

### 2.2. Analysis of the Mutation Landscape and Immune Microenvironment between the Different DDR Subtypes

The different mutation frequency and mutation landscape between DDR-subtype1 and DDR-subtype2 were characterized (Figure 3A,B). The analysis of driver genes showed that in the TCGA-PAAD cohort, patients with DDR-subtype1 showed higher mutation frequencies of KRAS (90.7%) and SMAD4 (30.8%) than patients with DDR-subtype2 (Figure 3C); patients with DDR-subtype2 were dominated by TP53 (83.8%) and CDKN2A (24.3%) mutations (Figure 3D). To explore the heterogeneity of the tumor immune microenvironment (TME) between DDR subtypes in PAAD, the differences in the components of immune cells and the expression of immune genes were analyzed. CIBERSORT analysis showed that DDR-subtype2 patients had significantly increased proportions of some immune cells (Figure 3E), such as activated memory CD4+ T cells and resting NK cells, and patients with DDR-subtype1 showed increased proportions of Tregs and activated NK cells. Antigen presentation and other immune functions also play a key role in the response to immunotherapy [10]. The expression levels of genes related to antigen processing and presentation, such as HLA-DOA, HLA-DPB2, HLA-DRA, HLA-DQA1, HLA-DRB5, HLA-DPA1 and HLA-DMB, were significantly higher in the DDR-subtype2 than the respective levels in the DDR-subtype1 (Figure 3F). In view of the significance differences in OS, somatic mutation and immune infiltration between two DDR subtypes, we desired to generate the specific signature of DDR-related gene expression profiles to render distinct the subtype of patients with PAAD and predict the prognosis.

### 2.3. Identification of Biological Processes Related to Differential Expression Genes (DEGs) of DDR Subtypes

The transcriptome differences between the two subtypes were investigated and 1081 DEGs (|Fold Change| > 1.5, FDR < 0.05, Appendix A) were screened from all 12,650 genes (Figure 4A,B). The gene set enrichment analysis (GSEA) revealed that DDR subtypes have distinct transcriptomic alterations. The top 20 enrichment pathways were demonstrated. Most of them were DDR-related, including “G2M Checkpoint”, “E2F Targets”, “Mitotic Spindle”, “UV Response DN” and “MTORC1 Signaling”. Some of enrichment pathways were immune-related, including “Inflammatory Response”, “IL6-JAK-STAT Signaling”, “Interferon-Gamma-Response” and “IL2-STAT5 Signaling” (Figure 4C), which suggested that the effect of different DDRs on the prognosis of PAAD may be related to the activation of the immune pathway.

### 2.4. Construction and Validation of DDR- and Immune-Based Prognostic Risk Score (DPRS) Model

Since all the above analyses were conducted in the population, it was not convenient to predict individual response patterns of DDR defects. To establish a promising biomarker, these DDR-based DEGs (*n* = 1081) were intersected with the immune genes (*n* = 2483, Appendix A) and altogether 191 overlapping genes were screened for subsequent analysis (Figure 5A). By univariate Cox regression analysis, 13 DEGs were found to be correlated with prognosis (Figure 5B). Finally, five genes (MET, ERAP2, AGER, TNFRSF4 and DMBT1) were selected for constructing the prognostic signature via least absolute shrinkage and selection operator (LASSO) regression and tenfold cross-validation (Figure 5C,D, Appendix A). Moreover, we explored the relationship between the five-gene signature and immune infiltration (Figure 5E). In addition, we found that the risk groups were consistent with the previously established DDR subtypes on the clustering heatmap of these five genes (Figure 5F); this partly confirmed the hypothesis that DDR and immunization may have an impact on prognosis of PAAD. High-risk group had remarkably decreased OS compared to the low-risk group in training set (Figure 6A,B), and similar results were also observed in the validation set (Figure 6E,F). The AUCs (area under the curve) were 0.784, 0.814 and 0.713 for 1-, 2-, and 3-year OS, respectively, indicating the reliability of the DPRS for predicting the outcomes of PAAD patients (Figure 6C,D). Furthermore, the prognostic efficiency of the DPRS was also validated in the GSE85916 cohort, with AUCs of 0.662, 0.617 and 0.554 for 1-, 2-, and 3-year OS (Figure 6G,H).

### 2.5. Correlation of DPRS Model with Tumor Mutation Burden (TMB) and Clinical Characteristics

Since high-risk patients showed significantly higher TMB in the training cohort (Figure 7A), we performed a further stratified analysis for TMB. As expected, high-risk patients with high TMB had the worst OS among all subtypes (Figure 7B,C). In patients with high TMB, the DPRS showed the capability to efficiently identify patients with a better survival benefit. We further investigated whether the DPRS had similar or better predictive validity to other clinical factors (Figure 7D,E). A nomogram (Figure 7F) was developed to predict the OS of patients including three independent prognostic factors (site, examined lymph nodes and risk scores). AUCs of the nomogram were 0.775, 0.851 and 0.721 for 1-, 2- and 3-year OS, respectively (Figure 7G). The calibration curve showed that mortality can be estimated accurately by the nomogram (Figure 7H). These results suggested that the DPRS model can be used both as an independent prognostic factor and in combination with clinical indicators.

### 2.6. Role of the DPRS in Predicting Therapeutic Benefits

The expression levels of immune checkpoints in the two groups were compared, and the CTLA4 and PDCD1 in the low-risk group were significantly up-regulated (Figure 8A,B). Following this, we predicted the sensitivity of poly-ADP-ribose polymerase (PARP, Olaparib and Niraparib), ATR (Berzosertib and AZD6738) and CHK1 (MK8776) inhibitors which are protein targets on the DDR pathway [11]. Patients with low-DPRS showed more sensitivity (IC50) to these inhibitors (all *p* < 0.05) (Figure 8C). The above results indicated that the DPRS might be of benefit in terms of efficiently predicting the current treatment for PAAD.

### 2.7. Heterogeneity of DDR-Based Signature among Cell Subtypes in PAAD

In order to further analyze the heterogeneity of DDR in a higher dimension, we obtained a transcriptome map of 41,986 cells in primary PAAD tumors using single-cell RNA-seq (scRNA-seq) analysis. We applied principal component analysis (PCA) and UMAP analysis on variably expressed genes across all cells and identified 13 main clusters including type1 ductal, type2 ductal, endothelial, endocrine, fibroblast, stellate, acinar, Ki67+ cell, CD8+ T cell, CD4+ T cell, B cells, plasmocyte and macrophage (Figure 8D). The marker genes for identifying cell types were demonstrated by dotplot in Appendix A. The results of single cell analysis showed that the expression of five genes were significantly increased in some specific clusters, especially in type2 ductal (Figure 8E). The Kruskal–Wallis (K-W) test also suggested that DPRS was significantly different among different cell types (Figure 8F).

### 2.8. Validation of Risk Genes for PAAD

In our five-gene model, two genes (MET and ERAP2) were associated with poor prognosis, and previous studies found that their expression was significantly elevated in PAAD compared to adjacent tissues. Therefore, we detected the expression of these two genes in a Chinese clinical cohort by immunohistochemistry (IHC, Figure 8G) and analyzed the impact of their expression levels on prognosis. Basic information about the cohort was presented in Appendix A. The results showed that patients with high expression of MET or ERAP2 had an inferior prognosis (*p* < 0.05, Figure 8H).

## 3. Discussion

There are four major driving mutations in PAAD, and unfortunately, there is no clinically applicable targeted therapy for these four genes [12]. In addition, there are mutated genes with low mutation frequency in other different pathways, such as the DDR genes, which contribute to tumor heterogeneity and individual differences between patients [13]. However, the mechanisms by which DDR genes promote tumorigenesis and therapeutic response were still not fully understood in PAAD. In this study, patients with PAAD were clustered into two subtypes based on the DDR genes and the differences between the two subtypes were discovered. A DDR- and immune-based risk-scoring model, DPRS, was constructed. The responses of different risk groups to possible therapeutic targets were further discussed.

Through a comprehensive analysis in this study, the genomic changes were significant between the two subtypes. KRAS and SMAD4 mutations were more common in DDR-subtype1, while TP53 and CDKN2A mutations were more common in patients with DDR-subtype2, who had poor prognosis. This was similar to an earlier classification of four subtypes; the ‘squamous’ subtype of pancreatic cancer was significantly higher frequency of TP53 mutations than the other subtypes and had inferior prognosis [3]. G1 regulates DNA damage repair as a cell cycle checkpoint, and loss of G1 checkpoint control is common in cancer through TP53, CDKN2A and ATM mutations [14]. SMAD4 is a central regulator of the transforming growth factor-β (TGF-β) signaling pathway, which plays an important role in tumor development by inducing angiogenesis and immunosuppression [15]. We also described the discrepancy of TME between two DDR subtypes; Tregs and activated NK cells were increased in DDR-subtype1. NK cells are cytotoxic lymphocytes of the innate immune system that are capable of killing cancerous cells [16]. Surprisingly, there were fewer Tregs in DDR-subtype2 than in subtype1, which was in line with a previous study that Treg depletion failed to relieve immunosuppression and led to accelerated tumor progression [17]. The DDR subtypes have different immune characteristics which indicated different immunotherapy responses between DDR subtypes. In general, DDR provides a new way to understand the regulatory mechanism of a tumor and its immune microenvironment. GSEA analysis further showed that our clustering strategy was feasible. The most abundant pathways focused on DDR- and immune-related signals, which suggested that the effect of different DDR subtypes on the OS of PAAD might be related to the activation of immune pathways.

To facilitate application, five DDR- and immune-based genes were finally selected and DPRS was subsequently constructed. Studies have consistently shown that MET overexpression is a negative prognostic indicator in a variety of malignancies, and activation of the HGF-MET axis was associated with the drug resistance of tumor cells [18,19,20]. In this study, MET was significantly negatively correlated with CD8+ T cells, CD4+ T memory activated cells, plasma cells and naïve B cells, respectively. Previous studies have shown that MET was significantly associated with the prognosis of immune “hot” and “cold” pancreatic cancer and the combined application of MET inhibition and PD-L1 blockade showed a significant therapeutic efficacy [21]. In our model, ERAP2 was another poor prognostic factor in addition to MET. ERAP2 might be a potential target for recoding epigenetic factors and enhancing the immunogenicity of malignant cells to develop an anticancer immune response. It has been furthermore shown to regulate endothelial cell cycle progression (G1/S conversion) by stimulating cyclin-dependent kinase (CDK)4/6 through VEGF [22], which also suggested that there was crosstalk between DDR and immune pathways.

In order to explore the underlying mechanism causing different OS between the high- and low-risk groups, three aspects including TMB, immune checkpoint and drug sensitivity were discussed. The real-world genomic characteristics of large-scale patients with pancreatic cancer showed that patients with somatic DDR gene mutations had higher TMB levels [23]. In this study, the DPRS model has the ability to distinguish high TMB and low TMB. In addition, our risk subgroups can be combined with TMB to significantly differentiate the OS of patients. Cancer immunotherapy is gaining momentum following the recent success of antibodies targeting checkpoint molecules CTLA-4 and PD-1 [24]. However, only a subset of patients responded to immune checkpoint blockade, especially in PAAD [25]. The loss of DNA repair may lead to an increased TMB and neoantigen burden that affects the response to immunotherapy [26]. Studies have shown that the expression of immune checkpoints can predict the efficacy of immunotherapy [27]. In this study, higher levels of immune checkpoint gene expression were observed in the low-risk group than that in the high-risk group, which suggested that the DPRS has the potential to guide immunotherapy. Furthermore, DDR dysfunction-related gene mutations could render PAAD susceptible to therapeutic interventions that increase the DNA damage load beyond tolerable thresholds such as PARP inhibitor-induced synthetic lethality [28], and preclinical trial results and data from ongoing clinical trials suggested a synergistic effect between PARP inhibitors and ICIs in solid tumors [29]. In this study, there were significant differences between the two risk groups in predicting sensitivity to PARP, ATR and CHK1 inhibitors. In addition, the single-cell data further suggested the heterogeneity of the DDR-based signature in different cell types. The score in type2 ductal cell was significantly increased, which was consistent with the results reported by the previous study that the type2 ductal cell was a subtype with more malignant features than type1 ductal cell [30].

There are limitations in our study. First, because this was a retrospective study, there was a lack of randomized trials of PAAD patients receiving immunotherapy and targeted therapy to verify the predictive performance of the model for treatment response. Second, our study focused on multi-cohort sequencing and clinicopathological data. Future in vivo and in vitro mechanism exploration may provide more information and validation for DDR subtype changes in PAAD and precision therapy.

## 4. Materials and Methods

### 4.1. Data Availability and Preprocessing

The messenger RNA (mRNA) expression matrix, harmonized to fragments per kilobase million (FPKM), and the related clinical information from patients with PAAD were extracted from the TCGA (https://portal.gdc.cancer.gov/projects/TCGA-PAAD/, accessed on 30 September 2021) database. We then merged the FPKM data for each patient into a matrix and converted the patient ID with the metadata file. The gene expression matrix used for downstream analysis is presented in Appendix A and related clinical data are provided in Appendix A. The mutation data of the included PDAC samples were downloaded from TCGA in the format of varscan2. The clinical data we downloaded contained 185 cases. Firstly, we removed 8 cases which histologically belong to “Neuroendocrine” and 1 “Discrepancy” to get 176 cases, and we then interleaved them with 178 cases with RNA-seq data (4 adjacent normal pancreas tissues were eliminated in 182 cases with expression data) to get 168 selected cases. The information of the final 168 patients we used is provided in Appendix A. For further verification, the clinical follow-up and gene chip datasets were obtained from the Gene Expression Omnibus (GEO, https://www.ncbi.nlm.nih.gov/geo/, accessed on 10 March 2022) database (GSE85916, *n* = 79). The raw expression matrix for single-cell transcriptome analysis and clinical characteristics of PAAD patients were obtained from CRA001160 in the Genome Sequence Archive (GSA, https://ngdc.cncb.ac.cn/gsa/browse/CRA001160, accessed on 1 March 2021) database.

### 4.2. DDR Genes-Based Subtyping

A total of 276 DDR genes were acquired from previous work based on MSigDB v5.0 and the knowledge-based curation of DDR pathways [9,31]. We performed unsupervised clustering with the transcriptomic profile of 276 DDR genes to identify subgroups of the TCGA cohort using the ‘ConsensusClusterPlus’ R package [32]. The following details were set for subgrouping: number of repetitions = 1000 bootstraps; pItem = 0.8 (resampling 80% of any sample); maxK = 7 (k-means clustering with up to 7 clusters). The ‘Nbclust’ R package [33] was further used to validate the optimal number of clusters for k-means clustering. The gene expression difference of DDR gene set between the two subtypes was demonstrated by R package ‘pheatmap’. The Kaplan–Meier (K-M) method with a log-rank test was performed to compare OS differences between the subgroups.

### 4.3. Characteristics for the DDR Subtypes

The mutation data of TCGA cohort were analyzed using the R package ‘maftools’. The waterfall plots were used to depict the mutation landscape between the two DDR subtypes. Based on RNA-seq expression matrix of TCGA cohort, the CIBERSORT algorithm was applied in analyzing the differences of immunocyte infiltration status between the two DDR subtypes with regard to 22 immunocyte subunits and 24 HLA genes. The immune profile differences between subtypes were then estimated by the Wilcoxon test.

### 4.4. Identification of DEGs between Different DDR Subtypes

The DEGs of DDR subtypes were identified by comparing gene transcription profiles of patients from the TCGA-PAAD database with the Wilcoxon rank-sum test, (|Fold Change| > 1.5, FDR < 0.05). The difference of DEGs after homogenization by setting ‘scale = row’ when using ‘pheatmap’ between the two subtypes was demonstrated by heatmap. We compared transcriptomic alterations between the DDR subtype1 and the DDR subtype2 by using GSEA. The hallmark gene sets were downloaded from http://www.gsea-msigdb.org/gsea/msigdb, (accessed on 9 March 2022). The GSEA analysis were performed based on the ‘ClusterProfiler’ R package.

### 4.5. Construction and Validation of Prognostic Signature Based on DDR and Immune Genes

In order to ensure the universality of the genes selected by us in different cohorts, we first performed log2(FPKM + 1) on the gene expression of TCGA and GEO cohorts, and then took their intersectant genes. Finally, the batch effect was removed by the ‘Combat’ function of the ‘sva’ R package and the gene expression matrices of the training and validation cohorts were output for subsequent analysis. The DEGs of DDR subtypes and immune hallmark genes (*n* = 2483) which were extracted from the immunology Database and Analysis Portal (ImmPort, https://www.immport.org/, accessed on 9 March 2022) were intersected. The intersectant genes were further analyzed by univariate Cox regression, and those with *p* value < 0.05 were considered as prognostic genes in the training cohort. These prognostic genes were then processed with the LASSO regression in order to avoid over-fitting and to delete those tightly correlated genes. Tenfold cross-validation was employed to select the minimal penalty term (λ). Following this, we established a signature for the PAAD patients. The risk score was established by the expression level of each gene multiplied by its corresponding regression coefficients derived from LASSO regression analysis of each gene, using the following formula:riskscore=∑Exp(mRNAi)×Coef(mRNAi)
where *mRNAi* is the *i*-th selected gene, and *Coef* is its regression coefficient. The risk scores were arranged in a sequence from low to high, and we took the median value to divide them into a high-risk group or low-risk group. The relationship between risk score model and previously constructed DDR subtypes was analyzed using the R package ‘pheatmap’. The K–M survival analysis was implemented to compare the OS of the two groups. The time-dependent receiver operating characteristic (ROC) analysis at 1, 2, and 3 years of prognostic value was used to assess the discrimination of the risk score in predicting the OS of PAAD using the R package ‘survivalROC’. A calibration curve was used to assess the agreement of predicted and observed values. External validation was conducted in the GSE85916 cohort.

### 4.6. Correlations among Risk Score, Molecular and Clinical Characteristics

The differences of TMB between low- and high-risk groups were estimated. The K-M method with a log-rank test was performed to compare OS differences between the groups. Uni- and multivariate Cox regressions were used to verify the prognostic role of the risk score and select clinical factors. A nomogram was then established using the R package ‘rms’ based on risk score and clinical factors with prognostic value (site and number of resected lymph node). The predictive effect of the nomogram was validated by ROC and calibration curve.

### 4.7. Prediction of Therapeutic Benefits in Patients with Distinct DPRS

The different expression of immune checkpoint genes between low- and high-risk groups were estimated. Using the ‘calcPhenotype’ function based on Genomics of Drug Sensitivity in Cancer (GDSC) database (www.cancerrxgene.org, accessed on 23 March 2022) in the “oncoPredict” R package, we entered the TCGA-PAAD cohort gene expression profiles of the two risk groups to predict the half-maximal inhibitory concentration (IC50) values of compounds or inhibitors [34]. The differences of IC50 between groups were estimated by the Wilcoxon test.

### 4.8. Single-Cell Analysis for DDR Heterogeneity Estimation

The scRNA-seq profiles were generated from 24 PAAD tumor samples, and their clinical information and metadata were included in Appendix A. For analysis of raw data (fastq) downloaded from GSA platform, the Cell Ranger Single-Cell toolkit (version 4.0.0) was used for alignment and barcode processing. The obtained filtered_feature_bc_matrix was used for subsequent analyses. For each sample, the ‘CreateSeuratObject’ function in ‘Seurat’ was used for quality control; we set cutoff min.cells to 3 and min.feature to 250. High-quality cells with a threshold of less than 20% mitochondrial gene expression were included in downstream analysis. Subsequently, all 24 samples were integrated according to sample ID using ‘merge’ function in R. We then normalized the data using the logarithmic transformation of the ‘NormalizeData’ function of ‘Seurat’. After normalization, 3000 highly variable genes were detected using the ‘FindVariableFeatures’ function in ‘Seurat’. PCA was used to reduce the noise of the variable gene matrix, and the first 50 components were selected for downstream analysis. We then applied ‘Harmony’ to minimize the batch impact of the individual. After initial quality control, we acquired single-cell transcriptomes in a total of 41,986 cells. The UMAP was generated by non-linear dimension reduction for visualization [35]. Using the ‘FindAllMarkers’ function of ‘Seurat’, marker genes for identifying cell types were obtained and demonstrated with dotplot. To explore our DDR-based signature heterogeneity in different cell types, the score of DDR-based signature in each cell was calculated and compared among different cell types. The ‘AddMouduleScore’ function was used to score gene sets, and the resulting score was the average expression of the calculated genes in each cell.

### 4.9. IHC

The tissue microarray for PAAD was boiled in citrate buffer (pH6.0) for 10 min for antigen repair and the IHC staining of MET (CST, #8198T, 1:300 dilution) and ERAP2 (FineTest, #FNab02826, 1:400 dilution) was performed based on the manufacturer’s protocol. Staining scores were graded at four levels by two independent experienced pathologists: negative (0), weak (1), neutral (2) and strong (3). All tissues were then divided into high (score ≤ 1) and low (score ≥ 2) groups.

### 4.10. Statistical Methods

Comparisons between two groups or more than two groups were conducted through the Wilcoxon test or K-W test, respectively. The chi-square test was used for analyzing the correlations between categorical variables. The correlation coefficient was calculated through Spearman analysis. All statistical analyses were performed using the R programming language (Version 4.1.0, R Core Team (2021). R: A language and environment for statistical computing. R Foundation for Statistical Computing, Vienna, Austria. URL https://www.R-project.org/). A difference of *p* < 0.05 indicated statistical significance unless specified otherwise.

## 5. Conclusions

Strategies for DDR subtype in PAAD and the novel risk-scoring model based on DDR and immune genes were constructed. This study provided information for the clinical management and decision making of patients with DDR heterogeneity. Our DDR-subtyping characterization contributed to a deeper understanding of the mechanisms associated with immunosuppression and poor prognosis in PAAD, and may provide new insights into the development of more effective biomarkers to predict therapeutic response in patients with PAAD.

## Figures and Tables

**Figure 1 ijms-23-10231-f001:**
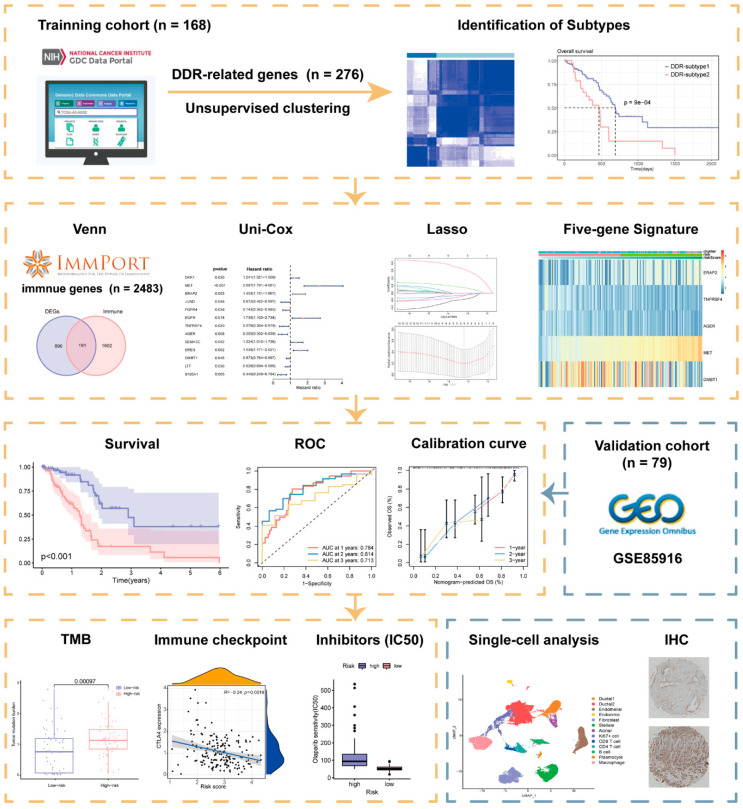
Workflow of the current study.

**Figure 2 ijms-23-10231-f002:**
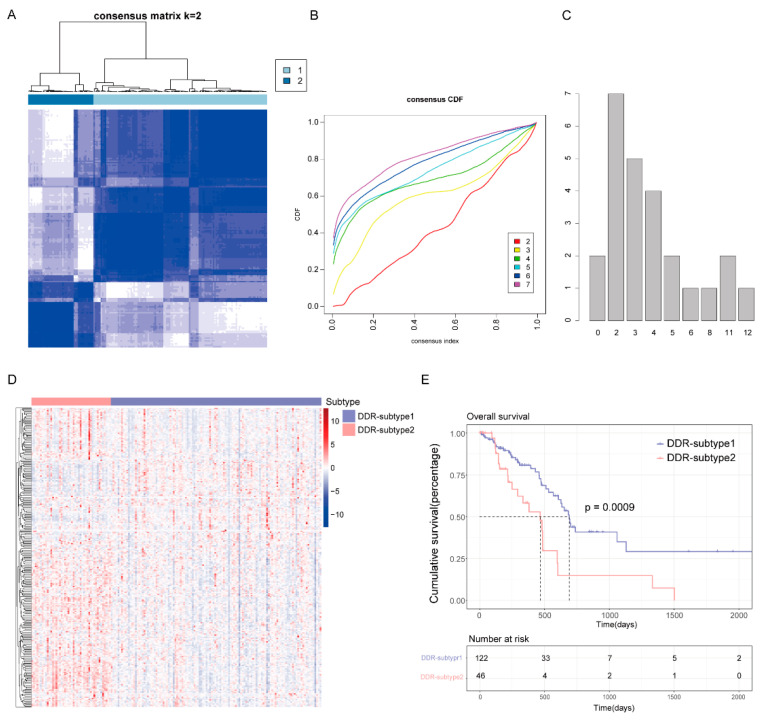
Consensus clustering for DNA damage repair (DDR) related genes in the TCGA-PAAD cohort (*n* = 168). (**A**) The consensus matrix of the DDR-related genes. (**B**) Cumulative distribution function (CDF) plot with consensus values ranging from 0 to 1. (**C**) The Nbclust plot represented the chosen optimal cluster number (k = 2) for DDR genes. (**D**) Heatmap of DDR-related genes in DDR subtypes. (**E**) Overall survival (OS) of patients in DDR-subtype1 and subtype2.

**Figure 3 ijms-23-10231-f003:**
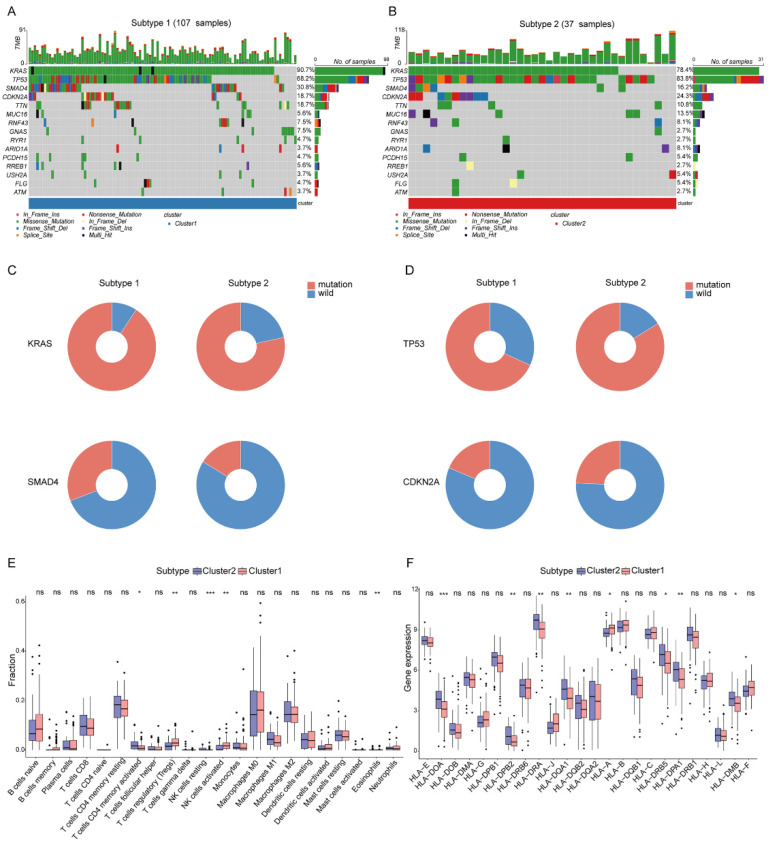
Somatic mutation and immune features between subgroups. Landscape of mutation profiles in DDR-subtype1 (**A**) and DDR-subtype2 (**B**). Mutation information of each gene in each sample is shown in the waterfall plot. Top panel shows individual tumor mutation burden. (**C**) Patients with DDR-subtype1 showed higher mutation frequencies of KRAS and SMAD4; (**D**) Patients with DDR-subtype2 showed higher mutation frequencies of TP53 and CDKN2A. Immune profile alterations (**E**) and human leukocyte antigen (**F**) between the DDR-subtypes. * represents *p* < 0.05, ** represents *p* < 0.01, *** represents *p* < 0.001, ns represents no significant difference.

**Figure 4 ijms-23-10231-f004:**
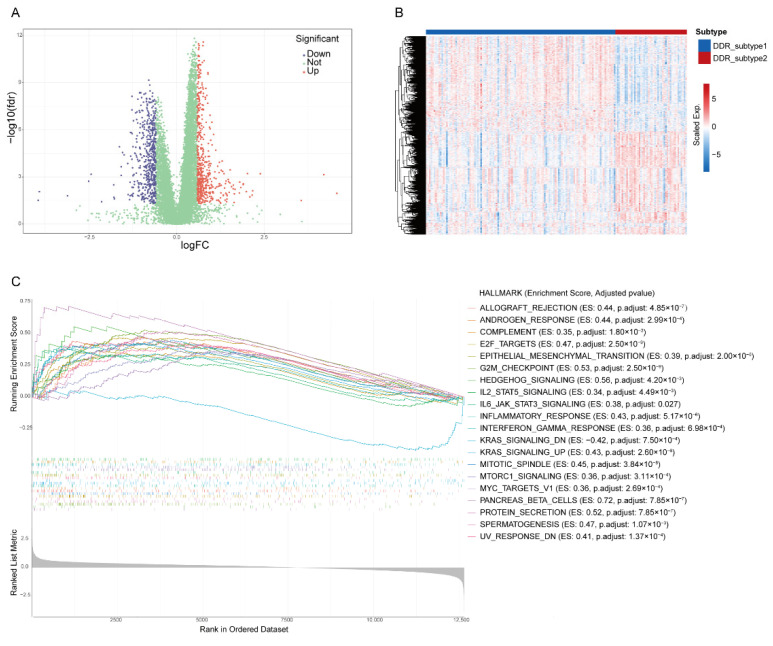
Identification of differential expression genes (DEGs) and related biological processes of DDR subtypes. Volcano plot (**A**) and heatmap (**B**) of DEGs in PAAD based on data from TCGA. (**C**) The top 20 hallmark gene sets of Gene Set Enrichment Analysis of DEGs.

**Figure 5 ijms-23-10231-f005:**
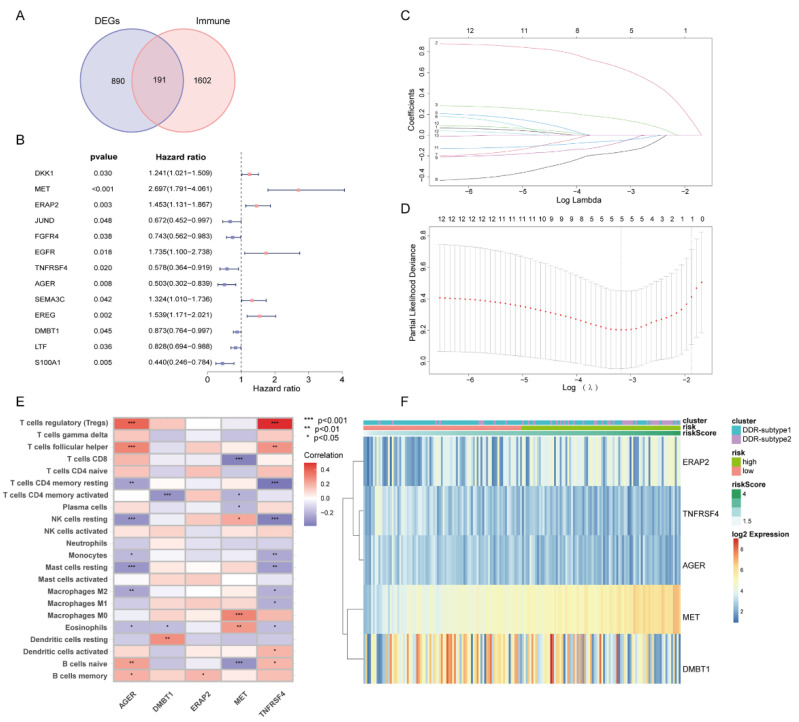
Generation of DDR- and immune-based signature. (**A**) Venn-diagram-described DDR-based DEGs (*n* = 1081) were intersected with the immune genes (*n* = 2483) and altogether 191 overlapping genes were screened. (**B**) Forest plot of univariate Cox regression analysis showed that 13 DEGs were associated with prognosis. (**C**) LASSO coefficient profiles. (**D**) Selection of the tuning parameter (lambda) in the LASSO model by 10-fold cross-validation based on minimum criteria for OS. (**E**) Heatmap of relationship between the 5-gene signature and immune infiltration. (**F**) Heatmap of 5-gene signature by unsupervised clustering. The DDR subtype, risk group and risk score as gene annotations were correlated.

**Figure 6 ijms-23-10231-f006:**
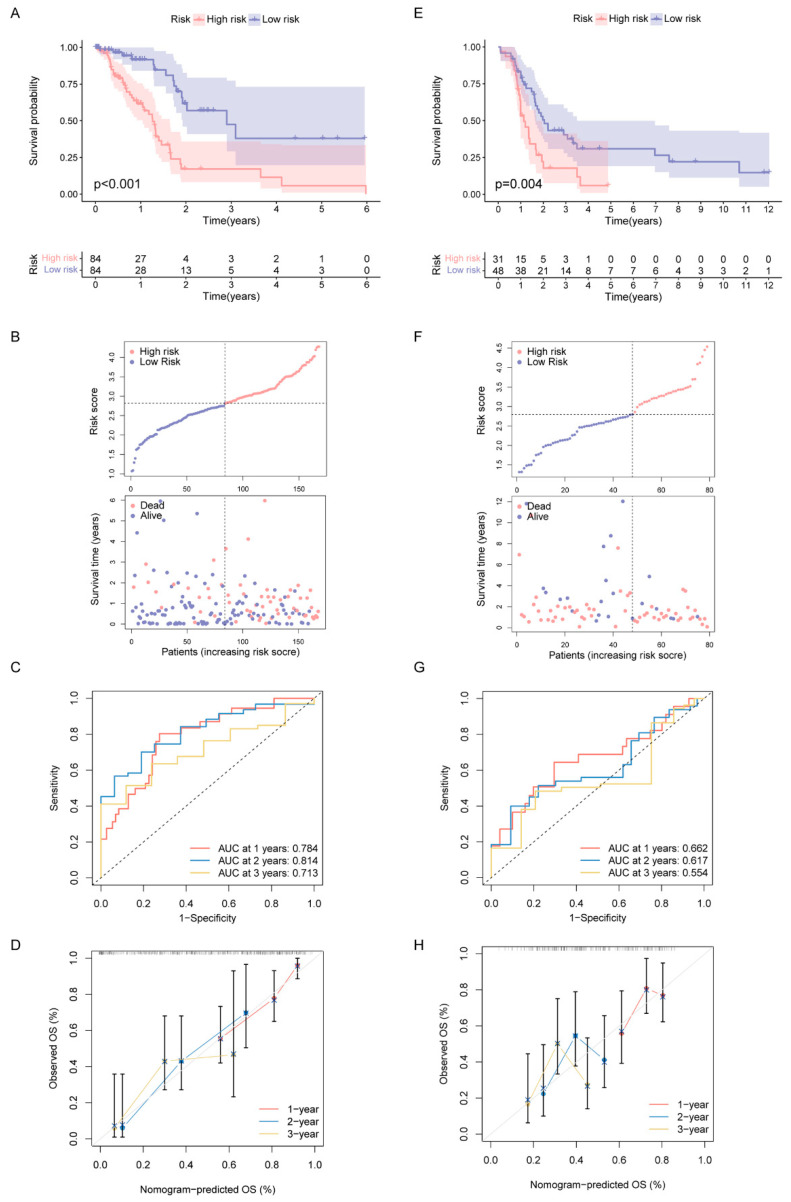
Construction and validation of DDR- and immune-based risk score (DPRS) model. (**A**–**D**) Construction and validation of TCGA training set. (**A**) The OS of training set. (**B**) Distribution of DPRS and OS of training set. Time-dependent ROC curves (**C**) and calibration curves (**D**) validation at 1, 2, and 3 years of prognostic value in TCGA cohort. (**E**–**H**) Construction and validation of GEO external validation set (GSE85916). (**E**) The OS of validation set. (**F**) Distribution of DPRS and OS of validation set. Time-dependent ROC curves (**G**) and calibration curves (**H**) validation at 1, 2, and 3 years of prognostic value in GEO cohort.

**Figure 7 ijms-23-10231-f007:**
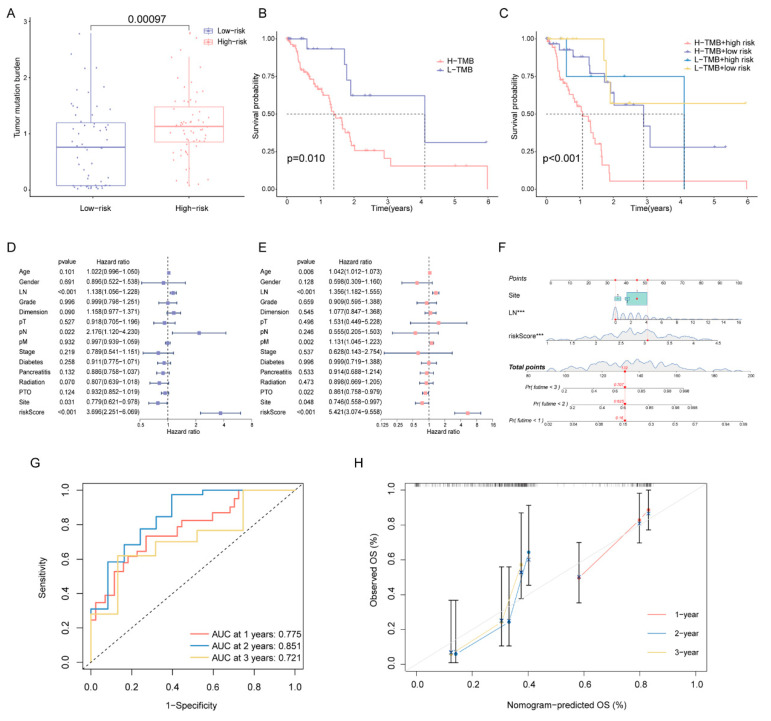
Correlation of risk score with TMB and clinical characteristics (**A**) Comparison of TMB between two DDR subtypes. (**B**) The OS of patients in the high- and low-risk groups. (**C**) The OS of patients in the risk groups combined with the TMB groups. (**D**) Univariate Cox analysis and (**E**) multivariate Cox analysis of clinical characteristics. (**F**) Nomogram predicting OS for PAAD patients. Time-dependent ROC curves (**G**) and calibration curves (**H**) validation at 1, 2, and 3 years of prognostic value in nomogram. *** represents *p* < 0.001.

**Figure 8 ijms-23-10231-f008:**
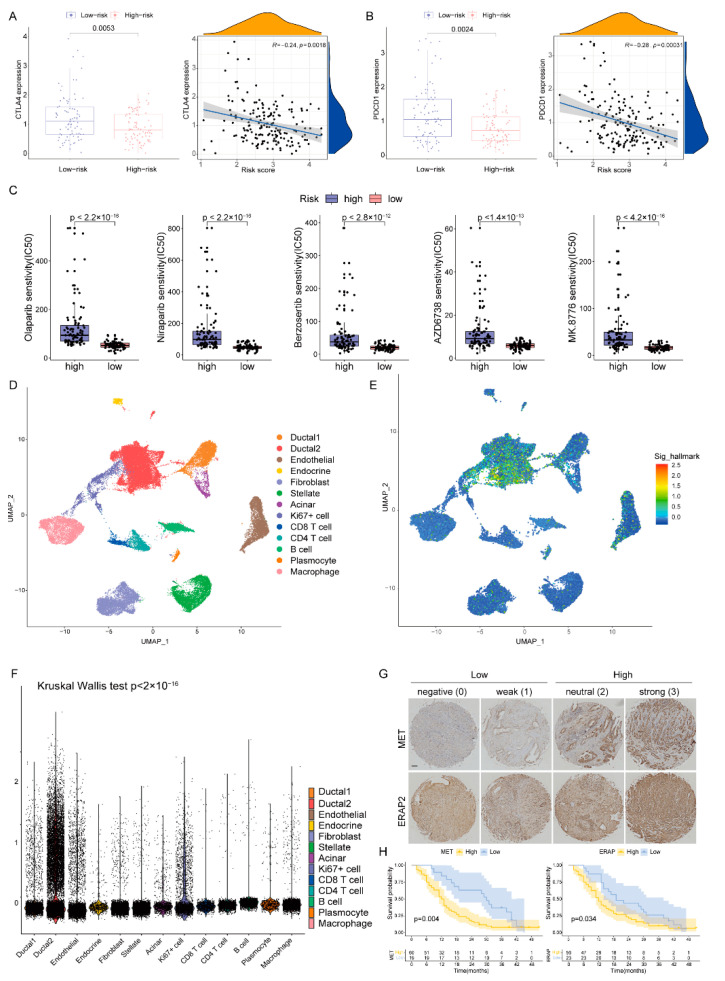
The boxplots and scatter plots of CTLA4 (**A**) and PDCD1 (**B**) in the low-and high-risk groups. (**C**) Boxplots showing estimated IC50 values for Olaparib, Niraparib, Berzosertib, AZD6738 and MK8776 in the TCGA-PAAD dataset (**D**) The uniform manifold approximation and projection (UMAP) plot demonstrates main cell types in PAAD. (**E**) The distribution of each type and DDR-based score expression in PAAD. (**F**) DDR-based scores in different cells are various (*p* < 0.05). (**G**) Staining scores of MET and ERAP2 were graded at four levels and two groups. Scale bar, 100 μm. (**H**) The OS of patients in the high and low expression of MET or ERAP2.

## Data Availability

The datasets analyzed during the current study are available in the TCGA, https://portal.gdc.cancer.gov/projects/TCGA-PAAD/, (accessed on 30 September 2021); GSE85916, https://www.ncbi.nlm.nih.gov/geo/query/acc.cgi?acc=GSE85916, (accessed on 10 March 2022); GSA, https://ngdc.cncb.ac.cn/gsa/browse/CRA001160, (accessed on 1 March 2021). Further inquiries can be directed to the corresponding authors.

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
