# Peer review of "Multiple Perspectives Reveal the Role of DNA Damage Repair Genes in the Molecular Classification and Prognosis of Pancreatic Adenocarcinoma"

_ijms, 2022, doi:10.3390/ijms231810231_

Round 1

Reviewer 1 Report

The study by Yjie Li et al used publicly available data to analysis the transcriptomic and mutational data from a set of pancreatic adenocarcinoma (PAAD) patients. With focus on DNA damage repair and immune genes to establish a 5-gene signature associated to patient overall survival capable of discriminate patient with high and low risk. The study globally was well conducted in terms of methodology in each step of the workflow. Although previous studies have already established numerous molecular signature for the classification of PAAD tumors, the focus on DDR and immune genes could be interesting. The study suffers of many issues and lack of information which are detailed in the comments below.

Author did not provide enough information about the expression datasets used for their analysis. All what we can read in the M&M section about data processing: The messenger RNA (mRNA) expression matrix harmonized to fragments per kilobase million and the related clinical information from patients with PAAD were extracted from TCGA. Indeed, they used public datasets but description of the nature of data and the preprecossing, analysis, downstream analysis is mandatory.

Authors should provide also metadata table of samples including patient IDs and related clinical data. Also, in the TCGA-PAAD dataset, there is 178 cases with available expression quantification data, how authors did select only 168? which patients have been filtered out and based on which criteria? Indeed, authors mentioned that 13 samples (9 non-PAAD + 4 adjacent normal..) were eliminated from the study, but it still not clear which patients have been used at the end.

Figure 4:

It is not clear how the differential expression analysis was performed. Which parameters were applied using “limma” R package?

To select DEGs authors used abs(fold change) > 1.5 and fdr<0.05. My concern is about using FC instead of log2FC, the threshold should be minimum log2FC > 1 which correspond to FC = 2. Applying a 1.5 fold leads to select genes with very low difference between the two groups. This is clearly visible on the figure 4A of the volcanoplot where most of the genes are close to log2FC =0.

At least authors should justify the use of these threshold and how this could alter results found here if they use log2FC>1.

  • The heatmap in figure 4C do not reflects the real range of expression levels. As presented in the figure, all expression values ranged from ~0.1 to |5| are presented by the same intensity of color (red or blue).

  • There is an issue with the final number of DEGs, in results section 2.3, authors wrote 1081 DEGs were screened from all 12650 DEGs. I do not understand what are the 12650 DEGs. It needs clarification.

  • GSEA results should give some statistics side by side with the enrichment score, the adjusted pvalue is required to avoid false positive, especially as we can see in the figure 4C, the enrichment scores look very low.

  • Authors statement: “The risk score was established by the expression level of each gene multiple its corresponding regression coefficients derived from LASSO regression analysis of each gene”, this is all we can read about how the score was established!

Not clear how the score of risk is calculated from cox univariate?

Not enough details, which expression value used?

Need the exact formula to calculate the risk score?

Figure 5F, color key missing for heatmap, values from 0 to ~10 correspond to what? Log2 expression?

Why is different from values in fig. 4B in which data seem to be centered around 0?

In fig 8C, additional information should be added such as the dots corresponding to patients in two groups. Also, number of samples in each group should be indicated. It seems that the low-risk group contains only a small number of samples (3 or 4 ?). In this case, statistics are very weak and conclusion from data is not very relevant.

In addition, in the Method section authors mentioned that they used cell lines to predict sensitivity and compare the two groups of patients. It is misleading or lacking of information.

Questions: how many cell lines used? what are the cell lines ID? the corresponding IC50 values?

Also, which expression data used to calculate the risk score for cell lines?

All these information are mandatory and are missing in supp data. It is not possible to review the data.

Even all these information have been provided, the conclusion made from this analysis is not applicable because the dose response results from cell lines could be very different compared to both pre-clinical models (such as organoids or PDX) and even more difficult to extrapolate to clinical setting for patients.

Single cell analysis (scRNA-seq) lacking many details and information. Metadata and clinical information should be provided in the supp material of the paper. It is not clear how the authors treated the single cell expression data, from raw expression matrix to plots. In the M&M section, they mentioned only “initial quality control”. More details needed to describe the analytic workflow. Which filters were applied? What were the criteria to keep or remove cells? Which kind of normalization applied? Again, how the risk score was calculated from single cell data? Authors should provide all data that allowed this calculation.

Reviewer 2 Report

I would like to congratulate the authors for the original manuscript 'Multiple perspectives reveal the role of DNA damage repair 2 genes in the molecular classification and prognosis of pancreatic adenocarcinoma". As the recent approval of PARP inhibitor (PARPi) olaparib for the treatment of metastatic, germline mutated BRCA1/2 (gBRCA) has shown that significant proportion of human PDACs with either somatic or germline mutations in DDR genes might benefit from targeted therapies. It is disappointing that there are no in vivo or in vitro experiments to validate some of the key findings of the paper. However the paper does provide vital novel information  for predicting the response to DDR-targeted inhibitors and immune checkpoint inhibitors (ICI) and it would be interesting to see if these findings are validated in further follow up studies.
